# Serum-Derived Neuronal Exosomal miRNAs as Biomarkers of Acute Severe Stress

**DOI:** 10.3390/ijms22189960

**Published:** 2021-09-15

**Authors:** Minkyoung Sung, Soo-Eun Sung, Kyung-Ku Kang, Joo-Hee Choi, Sijoon Lee, KilSoo Kim, Ju-Hyeon Lim, Gun Woo Lee, Hyo-Deog Rim, Byung-Soo Kim, Seunghee Won, Kyungmin Kim, Seoyoung Jang, Min-Soo Seo, Jungmin Woo

**Affiliations:** 1Department of Laboratory Animal Center, Daegu-Gyeongbuk Medical Innovation Foundation (DGMIF), Daegu 41061, Korea; tjdalsrud27@naver.com (M.S.); sesung@dgmif.re.kr (S.-E.S.); kangkk@dgmif.re.kr (K.-K.K.); cjh522@dgmif.re.kr (J.-H.C.); sjlee1013@dgmif.re.kr (S.L.); kskim728@knu.ac.kr (K.K.); 2Department of Psychiatry, School of Medicine, Kyungpook National University, Daegu 41944, Korea; hdrim@mail.knu.ac.kr (H.-D.R.); because99@hanmail.net (B.-S.K.); wonsh864@knu.ac.kr (S.W.); for-bluewish@hanmail.net (K.K.); seoyoung870314@daum.net (S.J.); 3Department of Veterinary Toxicology, College of Veterinary Medicine, Kyungpook National University, 80 Daehakro, Buk-gu, Daegu 41566, Korea; 4New Drug Development Center, Osong Medical Innovation Foundation, Chungbuk 28160, Korea; globaljh2019@gmail.com (J.-H.L.); gwlee1871@gmail.com (G.W.L.); 5Department of Orthopedic Surgery, Yeungnam University College of Medicine, Yeungnam University Medical Center, 170 Hyonchung-ro, Namgu, Daegu 42415, Korea

**Keywords:** acute severe stress, neuroinflammation, exosomes, miRNA, biomarkers

## Abstract

Stress is the physical and psychological tension felt by an individual while adapting to difficult situations. Stress is known to alter the expression of stress hormones and cause neuroinflammation in the brain. In this study, miRNAs in serum-derived neuronal exosomes (nEVs) were analyzed to determine whether differentially expressed miRNAs could be used as biomarkers of acute stress. Specifically, acute severe stress was induced in Sprague-Dawley rats via electric foot-shock treatment. In this acute severe-stress model, time-dependent changes in the expression levels of stress hormones and neuroinflammation-related markers were analyzed. In addition, nEVs were isolated from the serum of control mice and stressed mice at various time points to determine when brain damage was most prominent; this was found to be 7 days after foot shock. Next-generation sequencing was performed to compare neuronal exosomal miRNA at day 7 with the neuronal exosomal miRNA of the control group. From this analysis, 13 upregulated and 11 downregulated miRNAs were detected. These results show that specific miRNAs are differentially expressed in nEVs from an acute severe-stress animal model. Thus, this study provides novel insights into potential stress-related biomarkers.

## 1. Introduction

Stress is considered to be physical and psychological tension resulting from an adverse situation or environment to which an individual struggles to adapt and maintain homeostasis [1]. According to the type and severity of the stimulus, stress can not only disrupt homeostasis but may also cause various diseases and can even prove fatal [2]. Several studies have shown that stress is associated with diseases, such as ischemic stroke, cardiovascular disease, cancer, inflammatory bowel disease, and atopic dermatitis, as well with psychiatric disorders [3,4,5,6,7,8]. Other studies have indicated that neuroinflammation is exacerbated by the stress-activated hypothalamic-pituitary-adrenal (HPA) axis without direct brain damage [9]. Stress is also related to neurodegenerative diseases, such as Alzheimer’s disease and Parkinson’s disease [9]. 

The HPA axis is an important neuroendocrine system that moderates responses to psychological and physical stress and inflammation [6]. When an individual is stressed, a corticotropin-releasing hormone secreted by the hypothalamus stimulates the pituitary gland to release adrenocorticotropic hormone, which in turn stimulates the adrenal gland to release glucocorticoid hormones, i.e., effector hormones [10]. Glucocorticoid hormones, such as cortisol and corticosterone, are used as stress biomarkers. However, the most effective method by which stress is currently diagnosed clinically is through a combination of intensive interviews/surveys, glucocorticoid hormone measurements, and physiological measurements [11]. Although glucocorticoids are promising stress biomarkers, glucocorticoid levels can be affected by several factors, such as the circadian rhythm, sex, and stressor type [12,13,14]. Thus, it is difficult to quantify and analyze stress qualitatively using the expression patterns of these hormones. With the aim of overcoming such limitations, potential stress-related biomarkers were established, analyzed, and characterized in the present study.

Extracellular vesicles are classified into several types according to their size and biogenesis [15]. Among these types, exosomes are spherical, phospholipid, bilayered vesicles that are 30–200 nm in diameter and present in almost all body fluids, including cerebrospinal fluid, saliva, breast milk, and blood [16]. Exosomes are first secreted as vesicles from various cell types through a process of multivesicular endosome fusion with the plasma membrane, and they subsequently regulate intercellular communication [17]. These exosomes contain components, such as proteins, lipids, mRNA, and microRNAs (miRNAs), that are expressed in the cells of origin [17]. These components can exist in a stable state within the exosomes, which can themselves pass through the blood-brain barrier in both directions [18]. Thus, exosomes are being actively studied as potential diagnostic tools for many diseases including cancer, inflammatory bowel disease, and cardiovascular disease [19,20,21]. Additionally, exosomal miRNAs in cerebrospinal fluid have been studied as biomarkers for Parkinson’s disease and Alzheimer’s disease; the expression of miRNA in the exosomes of patients is known to differ from that in the exosomes of healthy controls [22,23].

One of the constituents of exosomes mentioned above, miRNAs, are small (17–24 nt), noncoding, endogenous RNAs that regulate post-transcriptional silencing of target mRNAs by binding to the 3′-untranslated region or open reading frame [24,25]. They can exist stably in various body fluids within exosomes and high-density lipoproteins or by binding to the argonaute2 protein [24]. Comparative analysis studies have revealed that miRNA expression patterns can be used as biomarkers in diseases, such as pancreatic cancer, allergic diseases, cardiovascular disease, and major depression [26,27,28,29].

In the current study, stress biomarkers were investigated by inducing stress in rats through the electric foot-shock method. Electric foot shock causes both physical and psychological stress and affects the subject animal both behaviorally and neurochemically [30]. Thus, this method is suitable for studying acute severe-stress-induced inflammation of the brain because it causes a significant stress response [30,31,32]. As advantages over other animal models of stress, the electric foot-shock model can be applied with a variety of intensities, durations, and frequencies [30,33].

Given that brain damage may be induced by stress [34], the expression of stress-specific biomarkers was measured following damage to the hippocampus in the stress-related animal model, while neuronal exosomes (nEVs) were isolated from the serum of these animals, and miRNA expression patterns were comparatively analyzed against the respective expression patterns of normal animals. Our findings suggest that neuronal exosomal miRNAs could be used as stress-specific biomarkers.

## 2. Results

### 2.1. Stress Hormone Levels Increased in the Acute Severe-Stress Group

Cortisol and corticosterone levels in serum were measured using an enzyme-linked immunosorbent assay (ELISA). Blood was collected at 0 h, 2 h, 4 h, 6 h, 9 h, 18 h, 20 h, 24 h (1 day), 72 h, (3 days), and 168 h (7 days) after stress induction. The acute severe-stress group exhibited significantly increased cortisol levels compared to those of the control group at all time points (Figure 1A: 0 h = 13.45 ± 4.91 ng/mL vs. control (4.21 ± 0.41), *p* = 0.00099, fold-change = 3.19; 2 h = 8.82 ± 2.45 vs. control (4.36 ± 0.85), *p* = 0.0018, fold-change = 2.02; 4 h = 9.77 ± 2.69 vs. control (4.56 ± 0.92), *p* = 0.0012, fold-change = 2.14; 6 h = 7.36 ± 1.83 vs. control (4.99 ± 0.85), *p* = 0.016, fold-change = 1.47; 9 h = 9.67 ± 2.25 vs. control (5.4 ± 1.33), *p* = 0.0025, fold-change = 1.79; 18 h = 6.37 ± 2.66 vs. control (2.51 ± 0.57), *p* = 0.0060, fold-change = 2.54; 20 h = 11.24 ± 4.39 vs. control (4.71 ± 1.29), *p* = 0.0058, fold-change = 2.39; 1 day = 10.08 ± 4.42 vs. control (3.37 ± 0.65), *p* = 0.0043, fold-change = 2.99; 3 days = 5.91 ± 1.45 vs. control (3.9 ± 1.27), *p* = 0.029, fold-change = 1.52; 7 days = 5.58 ± 1.3 vs. control (3.35 ± 0.81), *p* = 0.0051, fold-change = 1.67). 

Significant increases in corticosterone were observed at 0 h and 1 day after exposure to stress, and a significant decrease was confirmed at 6 h after stress induction (Figure 1B: 0 h = 224.31 ± 93.41 ng/mL vs. control (82.88 ± 62.32), *p* = 0.012, fold-change = 2.71; 6 h = 34.24 ± 17.22 vs. control (105.91 ± 68.10), *p* = 0.031, fold-change = 0.30; 1 day = 119.58 ± 80.09 vs. control (19.73 ± 13.38), *p* = 0.013, fold-change = 6.061). At all other time points, corticosterone levels were not significantly different from those in the controls (Figure 2B: 2 h = 45.66 ± 31.58 vs. control (75.28 ± 38.15), *p* = 0.174, fold-change = 0.61; 4 h = 106.94 ± 45.64 vs. control (94.71 ± 68.77), *p* = 0.726, fold-change = 1.13; 9 h = 100.79 ± 72.11 vs. control (94.66 ± 60.06), *p* = 0.876, fold-change = 1.064; 18 h = 36.08 ± 32.38 vs. control (20.84 ± 19.65), *p* = 0.348, fold-change = 1.73; 20 h = 134.43 ± 75.67 vs. control (87.40 ± 76.77), *p* = 0.310, fold-change = 1.78; 3 days = 22.51 ± 22.17 vs. control (68.79 ± 73.10), *p* = 0.169, fold-change = 0.33; 7 days = 85.69 ± 76.35 vs. control (65.22 ± 35.14), *p* = 0.564, fold-change = 1.31). 

### 2.2. Pathological Analysis of Neuroinflammation Marker Levels in the Hippocampus of the Acute Severe-Stress Group

Acute severe-stress-induced changes in the levels of neuroinflammation marker proteins were detected following hematoxylin and eosin (H&E) staining and immunohistochemistry (IHC). According to H&E staining, there were no significant differences in the hippocampuses of stress-exposed rats and control rats at 1, 3, and 7 days after severe-stress induction (Figure 2A–D). IHC staining and quantification were performed to confirm the expression levels of neuron-associated inflammation markers, i.e., brain-derived neurotrophic factor (BDNF), cyclooxygenase-2 (COX2), and glial fibrillary acidic protein (GFAP) (Figure 2E–P). BDNF expression levels in the hippocampus did not differ across time points (Figure 3A: Day 1 = 1.75 ± 0.061 related to control, *p* = 0.051; Day 3 = 0.51 ± 0.055 related to control, *p* = 0.12; Day 7 = 1.45 ± 0.084 related to control, *p* = 0.16). However, COX2 expression levels were significantly upregulated at 1, 3, and 7 days after stress induction (Figure 3B: Day 1 = 1.66 ± 0.16, *p* = 0.013 related to control; Day 3 = 1.40 ± 0.052 related to control, *p* = 0.0041 related to control; Day 7 = 2.31 ± 0.34 related to control, *p* = 0.013). Similarly, GFAP expression levels were significantly altered at 1, 3, and 7 days after exposure to stress (Figure 3C: Day 1 = 1.51 ± 0.14 related to control, *p* = 0.021; Day 3 = 1.59 ± 0.16 related to control, *p* = 0.017; Day 7 = 1.92 ± 0.17 related to control, *p* = 0.0041).

### 2.3. Characterization of Total Exosomes (tEVs) Isolated from Serum

EVs were initially isolated to determine the time at which brain damage was most prominent, which was 7 days after electric foot shock. Before the separation of the nEVs, the characteristics of the tEVs isolated from serum were confirmed using several methods. First, transmission electron microscopy (TEM) was used to confirm their shape and size; images showed that the exosomes isolated from serum had spherical bilayer membranes and were 30–200 nm in diameter (Figure 4A). Next, nanoparticle tracking analysis (NTA) was conducted to validate the size of the exosomes, which were confirmed to be 30–200 nm in diameter (Figure 4B). Finally, flow cytometry (FACS) was performed to identify and quantify the presence of CD9 and CD63, which are exosome markers; results showed that 98.96% and 78.36% of the particles present in the exosomes were CD9-positive and CD63-positive, respectively (Figure 4C).

### 2.4. Identification of Serum-Derived nEVs 

The nEVs were isolated using an anti-L1 cell adhesion molecule (L1CAM; i.e., CD171) biotinylated antibody for immunoprecipitation (Figure 5A). Western blotting was then used to characterize the nEVs. Results showed that the nEV marker CD171 and neuronal markers, such as neuron-specific class III beta-tubulin (TUJ1), neuron-specific enolase (NSE), and neuronal nuclei (NeuN) were present in the nEVs isolated from serum. On the other hand, these markers were rarely identified in the tEVs. Additionally, tumor susceptibility gene 101 (TSG101) and β-actin were confirmed in both the tEVs and nEVs (Figure 5B).

### 2.5. Changes in Exosomal miRNA Expression Induced by Acute Severe Stress

Next-generation sequencing (NGS) was performed to confirm that miRNA expression patterns were altered by stress. We conducted NGS on the control group and the Day 7 group, which showed the greatest change in the expression of neuroinflammatory markers in the hippocampus. Results confirmed that 13 miRNAs were upregulated and 11 miRNAs were downregulated in the stress group relative to the control group (Figure 6). In the stress group, the upregulated miRNAs were as follows: let-7a-1-3p, let-7a-5p, let-7b-3p, let-7b-5p, let-7c-2-3p, let-7c-5p, let-7d-3p, let-7f-1-3p, let-7f-5p, miR-126a-5p, miR-3473, miR-466b-3p, and miR-98-3p. The downregulated miRNAs in the stress group were as follows: let-7d-5p, let-7g-5p, let-7i-5p, miR-140-3p, miR-17-5p, miR-191a-5p, miR-19b-3p, miR-24-3p, miR-30c-5p, miR-425-5p, and miR-93-5p (Table 1, with sequences in Appendix A).

### 2.6. Prediction of Target Genes

We predicted the miRNA target genes using the mirWalk. The target gene prediction results are shown in Appendix A. Gene ontology (GO) analysis was performed to confirm that the genes differentially expressed under acute severe stress were associated with neurons. Analysis of each biological process (BP), cell component (CC), and molecular function (MF) showed the 10 most enriched GO terms (Figure 7A–C). Kyoto Encyclopedia of Genes and Genomes (KEGG) pathway was then performed to identify that expression under acute severe stress was associated with neuron-related pathways. As a result, it can be seen that it is related to neuron-related signaling pathways (Figure 7D). 

## 3. Discussion

The objective of this study was to identify and analyze indicators expressed in a specific stress situation that could potentially be used as stress-related biomarkers. nEVs were isolated from the sera of electric foot-shock-induced acute severe-stress model rats, and the expression patterns of miRNAs contained in the exosomes were compared with those of unstressed animals. This model was used as previous studies have shown that electric foot shock causes activation of the HPA axis and psychiatric disorders, such as anxiety and post-traumatic stress disorder, in animals [32,35,36].

To determine whether electric foot shock caused stress as anticipated, glucocorticoids were analyzed in model rats; the concentrations of cortisol and corticosterone in serum were increased in shocked animals. These results are in agreement with those of other studies of rodents demonstrating that electric foot shock induces acute psychological stress and increases the secretion of corticosterone through the HPA axis [37,38]. Although corticosterone is the main upregulated stress-related hormone in rodents, an earlier study showed that cortisol was upregulated following electric foot shock; indeed, corticosterone levels were rapidly reduced compared with cortisol levels after incremental induction by several stressful states [35].

Changes in the expression patterns of neuron-related markers, such as BDNF, COX2, and GFAP, in the hippocampus were confirmed in the acute severe-stress model. BDNF is involved in processes related to learning and hippocampal memory, as well as the maintenance, survival, and plasticity of neurons [36,39]. Changes in BDNF expression have been identified in studies of conditions such as stress, psychiatric disorders, and neurodegenerative disorders [39,40]. In the present study, BDNF expression in the acute severe-stress model increased compared with that in the normal animals at the initial stage of stress induction and then decreased rapidly. However, these changes were not statistically significant. It has been reported that when neuroinflammation is induced with lipopolysaccharides, the expression pattern of BDNF is similar [41]. Additionally, a previous study has shown that BDNF initially decreases when acute stress is induced in rats but that this decrease does not persist [42].

The expression of COX2 and GFAP were significantly increased in the hippocampuses of acute severe-stress model rats. COX2 is known to be involved in cytokine-induced depression and sickness behavior [43]. It is also a prostaglandin-endoperoxide synthase and a marker of neural damage in the injured brain [44]. Furthermore, increased expression of COX2 can be a major factor in brain injury and stress-induced neuroinflammation [45,46]. GFAP is a marker of astrocytes, which are critical regulators of neuroinflammation in the central nervous system [47]. In previous studies, similar to the present study, GFAP was upregulated following the induction of acute severe stress [48,49]. Taken together, these results confirm that the acute severe-stress model used in this study exhibits a phenotype suitable for stress-related studies. 

The possibility of using exosomes as disease-specific biomarkers in various diseases has previously been suggested by other researchers. In the current study, exosomes from acute severe-stress model rats were isolated and analyzed with results being confirmed by TEM and NTA. In addition, FACS analysis showed the presence of the traditional exosome markers CD9 and CD63 in the isolated exosomes. Previous studies have identified exosomes using tetraspanins such as CD9, CD63, and CD81 (which organize membrane microdomains), along with TSG101 (which is involved in multivesicular endosome biogenesis) [25,50,51]. 

Because stress is closely related to brain damage, nEVs were also isolated and characterized here. After exosomes are isolated from serum, immunoprecipitation of CD171 is typically used to isolate nEVs [52,53]. CD171, also known as L1CAM, is expressed in nervous tissue and some cancer cells; it is known as a cell surface antigen in the central nervous system [54,55] and plays an important role in neural development [55]. After the isolation of tEVs from serum, nEVs were isolated by immunoprecipitation using biotinylated antibodies against the neuronal surface marker CD171, as described in previous studies [52]. Consistent with previous studies, the nEVs isolated in the present work included neuronal markers specifically expressed in neurons as well as CD171 [52,53], i.e., TUJ1, NSE, and NeuN were also identified as neuronal markers [50,51,56]. 

Many studies have investigated exosomes as biomarkers for diseases. In particular, miRNAs and proteins in exosomes have been studied using bioinformatics [57,58]. Thus, in the current study, NGS was performed to identify differentially expressed miRNAs in nEVs following electric foot shock. Expression patterns of miRNAs in the nEVs differed between stressed and unstressed animals. Differently expressed miRNAs in the stressed state were identified in the miRDB database (miRDB.org) and found to target several genes, including some neuron-related genes. For example, let-7b suppresses the proliferation of neural stem cells and accelerates the differentiation of neural stem cells in the mammalian brain [59]. In addition, let-7a-1-3p regulates the function of microglia, i.e., immune cells in the brain associated with BDNF signaling [60,61]. Moreover, several of the miRNAs identified are associated with psychological disorders such as major depressive disorder. Previous studies have confirmed that let-7a-5p, let-7c-5p, and let-7i-5p are upregulated in acute social defeat stress and chronic unpredictable mild stress (CUMS)-induced depression [62,63]. Here, let-7a-5p and let-7c-5p were upregulated, whereas let-7i-5p was downregulated. Additionally, let-7b-5p and miR-126a-5p, which are upregulated in depression, were upregulated in the acute severe-stress model rats, whereas miR-140-3p, which is downregulated in repeated social defeat stress and CUMS-induced depression, was downregulated in these animals [62,63]. These data confirm that some miRNAs in nEVs were affected by stress and that these miRNAs could be used as biomarkers of both stress and stress-induced psychiatric disorders.

In addition, we analyzed genes targeted by neuronal exosomal miRNAs with significantly altered expression. As a result, it has been shown to regulate a variety of genes, including neuron-related genes. For example, Forkhead Box O (FoxO) deficiency is known to affect the development of neurons [64]. There are also studies confirming that oxytocin, a neuropeptide, is related to the development of neuronal precursor cells in the brain [65]. Additionally, oxytocin was associated with cortisol level in psychological stress [66,67]. However, the genes analyzed are diverse and further studies in other stressful situations must be performed to find genes associated with stress.

In summary, nEVs were successfully isolated from a stress-related animal model and analyzed to confirm specific changes in miRNA expression compared with the respective expression in unstressed animals. Consequently, biomarkers related to stress were potentially identified. However, stress-related diseases are diverse and the causes, types, and phenotypes of disease can differ. Therefore, it will be necessary to conduct further research into various stress-related conditions, such as inflammatory colitis, neuropathic pain, and noise-induced hearing loss, to further study and validate the expression of these candidate stress-related biomarkers. Additionally, only miRNAs in nEVs were analyzed in this study, but other components, such as mRNAs, proteins, and lipids, need to be analyzed in further studies.

## 4. Materials and Methods

### 4.1. Animals

Eight-week-old male Sprague-Dawley rats were housed under controlled temperature, light, and humidity conditions (24 ± 2 °C, 12/12 h light/dark cycle with lights on at 07:00, and 50% ± 20% relative humidity). Additionally, animals had ad libitum access to food and water. All animal experiment protocols were reviewed and approved by the Institutional Animal Care and Use Committee of KPC Co., Ltd., Kyunggido, Korea (KPC-IACUC; approval no. P181112, 14 February 2018) and were conducted in accordance with their guidelines.

### 4.2. Acute Severe-Stress Model Protocol

Animals were divided, with six animals in each group. Acute severe stress was induced by electric foot shock. Stress groups were exposed to the foot-shock test for 20 min (8-s shock duration at 5-min intervals; four shocks in total). To measure stress-related hormone levels using ELISA, blood was collected at 0 h, 2 h, 4 h, 6 h, 9 h, 18 h, 20 h, 24 h (1 day), 72 h (3 days), and 168 h (7 days) after stress induction. Cortisol and corticosterone were measured using a Cortisol ELISA Kit (FineTest, Wuhan, China) and Corticosterone ELISA Kit (Enzo Life Sciences, Ann Arbor, MI, USA) according to the manufacturers’ instructions.

In addition, Day 1, Day 3, and Day 7 groups were sacrificed at 1 day, 3 days, and 7 days after stress induction for histopathological analysis and exosomes isolation (Figure 8). At these respective times, all animals were anesthetized with 3% isoflurane (Hana Pharm, Seoul, Korea) to minimize pain, and then 1.5 mL blood was collected from the abdominal vein.

### 4.3. Histopathology

Brain samples were fixed in 10% neutral buffered formalin. Following fixation, the tissues were dehydrated, cleared, and embedded in paraffin. Sections from the brain paraffin blocks were cut to a thickness of 4 μm. H&E staining was performed using Dako CoverStainer (Agilent, Santa Clara, CA, USA). To confirm the expression of neuroinflammation marker proteins, IHC was performed. The brain sections were stained with antibodies against BDNF, COX2, and GFAP (1:500; Abcam, Cambridge, UK) primary antibodies, and the labeled polymer Dako EnVision + System-HRP (Agilent) was used according with the manufacturer’s instructions. After staining, the brain sections were scanned with a Pannoramic SCAN II scanner (3DHISTECH Kft., Budapest, Hungary). Neuroinflammation markers in the hippocampus were quantified using ImageJ software version 1.53a (NIH, Bethesda, MD, USA).

### 4.4. Isolation of tEVs and nEVs

tEVs were isolated from serum using ExoQuick solution (System Bioscience, Palo Alto, CA, USA) with some modifications to the manufacturer’s instructions. After centrifugation to remove cell debris, the serum of six animals in each group were pooled. A total of 2 mL of serum and 500 μL of solution were mixed. After the mixture was centrifuged, the pellet was resuspended in 200 μL phosphate-buffered saline (PBS).

nEVs were isolated as described by Mustapic et al. [52] with some modifications. Exosomes isolated from serum were incubated with anti-CD171 (L1CAM; Bioss Antibodies, Beijing, China) antibody for 1 h at 4 °C on a rotating mixer. After adding Pierce Streptavidin Plus Ultralink Resin (Thermo Fisher Scientific, Waltham, USA) and PBS, the samples were again incubated for 1 h at 4 °C on a rotating mixer. The samples were then pelleted by centrifugation at 200× *g* for 10 min at 4 °C. The supernatants were removed from the samples and the pellets were resuspended in 200 μL 0.1-M glycine-HCl (Biosesang, Seongnam, Korea). After mixing for 10 s and vortexing for 30 s, the samples were pelleted by centrifugation at 4500× *g* for 10 min at 4 °C. Finally, the supernatants were transferred to new tubes before Tris-HCl (Biosesang) and PBS were added.

### 4.5. Flow Cytometry (FACS)

Exosomes isolated from serum were incubated with aldehyde/sulfate latex beads (Invitrogen, Carlsbad, CA, USA) for 15 min at room temperature (RT). PBS supplemented with 3% bovine serum albumin was then added and the samples were incubated overnight on a rotating mixer. The bead-coupled exosomes were pelleted by centrifugation at 3000× *g* for 10 min and washed with PBS. The samples were then pelleted by further centrifugation at 3000× *g* for 10 min. Subsequently, the supernatants were removed from the samples and the pellets were resuspended in PBS containing anti-CD9 and anti-CD63 antibodies (BioLegend, San Diego, CA, USA) for 1 h at RT. Afterwards, the samples were again pelleted by centrifugation at 3000× *g* for 10 min before the pellets were resuspended in PBS. Exosome markers were detected using flow cytometry (Galios, Beckman Coulter, Brea, CA, USA). Following detection, analysis was performed with Kaluza analysis software (Beckman Coulter, Brea, CA, USA).

### 4.6. TEM

Exosomes isolated from serum were resuspended in cold distilled water. These exosome suspensions were loaded on Formvar carbon-coated grids (Ted Pella, Inc., Redding, CA, USA) and fixed in 2% paraformaldehyde for 10 min. The solution was then removed and the samples were dried. Grids were observed by bioTEM (Hitachi HT7700) (Hitachi, Chiyoda, Tokyo, Japan).

### 4.7. NTA

NTA was performed using a Nanosight NS300 (Malvern, UK) according to the manufacturer’s instructions.

### 4.8. Western Blot

Western blotting was used to identify the exosome marker (TSG101), nEV marker (CD171), and neuronal markers (TUJ1, NSE, and NeuN). For lysis, M-PER and Halt Protease and Phosphatase Inhibitor Cocktail (Thermo Fisher Scientific) were added to each tEV and nEV sample isolated from serum. Sample concentrations were measured using a Pierce BCA Protein Assay Kit (Thermo Fisher Scientific) according to the manufacturer’s instructions. Twenty micrograms of exosome lysates were loaded on Bolt 4–12% Bis-Tris Plus Gels (Invitrogen, Carlsbad, CA, USA) and transferred to polyvinylidene fluoride membranes (Invitrogen). The membranes were then blocked with TBS-T supplemented with 5% skim milk for 1.5 h at RT. Following blocking, the membranes were incubated with TSG101 (1:1000; Novus Bio, Littleton, CO, USA), CD171, β-actin (1:1000; Santa Cruz Biotechnology, Inc., Santa Cruz, CA, USA), TUJ1, NeuN, and NSE (1:1000; Abcam, Cambridge, UK) primary antibodies overnight at 4 °C. After the membranes were washed with TBS-T, they were incubated with horseradish peroxidase-conjugated secondary antibody (diluted 1:2000) for 1 h at RT. Following the reaction, the membranes were again washed with TBS-T. The bands were developed using EzWestLumi Plus (ATTO, Tokyo, Japan) and analyzed using ImageQuant LAS 4000 (GE Healthcare, Buckinghamshire, UK). Western blot was performed in triplicate.

### 4.9. NGS for Exosomal miRNA Analysis

We performed NGS using neuronal exosomes isolated from the pooled serum of each Day 7 group (*n* = 6) and control group (*n* = 6), which showed the greatest difference in expression of neuroinflammatory markers in the hippocampus. Exosomal smRNA isolation and library preparation were performed by Macrogen (Seoul, Korea) using the SMARTer smRNA-Seq Kit (Clontech Laboratories, Inc., Mountain View, CA, USA) according to the manufacturer’s instructions. Subsequently, miRNA sequencing was conducted by Macrogen using the HiSeq 2500 system following the HiSeq 2500 System User Guide Document #15035786 v02 HCS 2.2.70. Differentially expressed miRNAs were identified with a threshold *p* < 0.05.

GO analysis was performed to analyze the functional enrichment of differentially expressed miRNAs, and KEGG pathway analysis was performed to identify significantly enriched signaling pathways. We used mirWalk to predict miRNA target genes that showed significant expression changes by stress, and then performed analysis. All the enrichment analysis was conducted using the database for annotation, visualization, and integrated discovery (DAVID) v6.8 (http://david.abcc.ncifcrf.gov/) (Access date: 20 May 2021). *** *p* < 0.001 was applied as the criterion.

### 4.10. Statistical Analysis

All data were statistically analyzed using Student’s *t* test. The level of statistical significance was set at *p* < 0.05. The statistical data were analyzed by Microsoft Excel software.

## Figures and Tables

**Figure 1 ijms-22-09960-f001:**
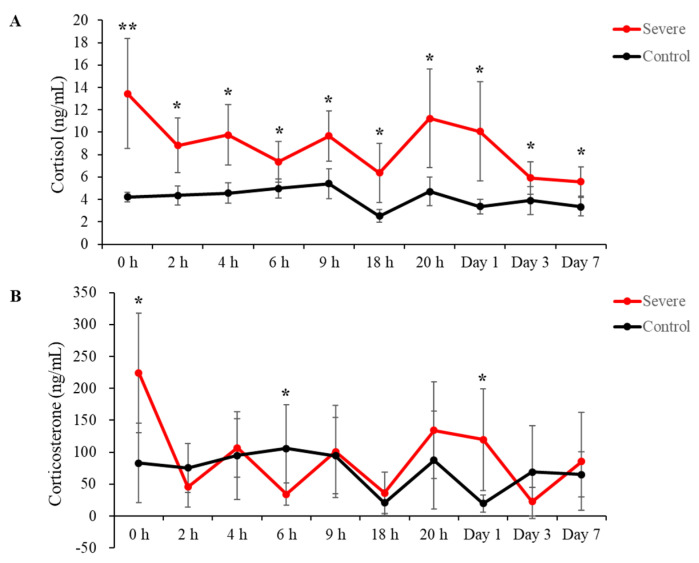
The hypothalamic-pituitary-adrenal (HPA) axis was influenced by acute severe stress. (**A**) Changes in cortisol concentrations induced by acute severe stress. (**B**) Alterations in corticosterone caused by acute severe stress. All data are expressed as means ± SD; * *p* < 0.05 and ** *p* < 0.001, *n* = 6 per group.

**Figure 2 ijms-22-09960-f002:**
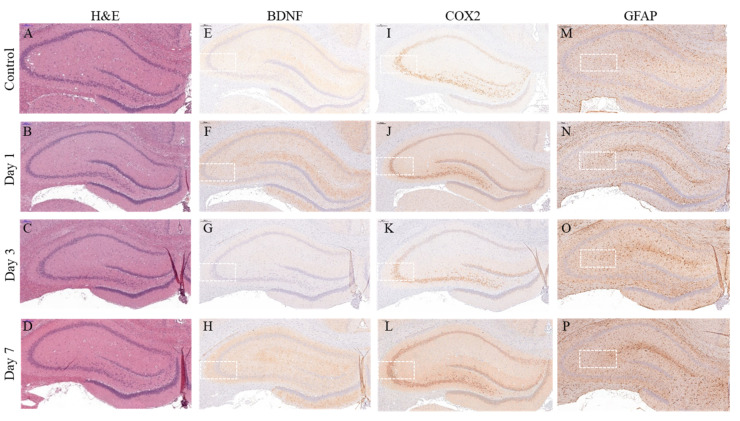
Acute severe stress led to changes in the expression patterns of neuroinflammation markers in the hippocampus. Representative hematoxylin and eosin (H&E) staining images of the hippocampus: (**A**) Control; (**B**) Day 1; (**C**) Day 3; and (**D**) Day 7. Images of immunohistochemical (IHC) staining of brain-derived neurotrophic factor (BDNF) in the hippocampus: (**E**) Control; (**F**) Day 1; (**G**) Day 3; and (**H**) Day 7. Images of IHC staining of cyclooxygenase-2 (COX2) in the hippocampus: (**I**) Control; (**J**) Day 1; (**K**) Day 3; and (**L**) Day 7. Images of IHC staining of glial fibrillary acidic protein (GFAP) in the hippocampus: (**M**) Control; (**N**) Day 1; (**O**) Day 3; and (**P**) Day 7.

**Figure 3 ijms-22-09960-f003:**
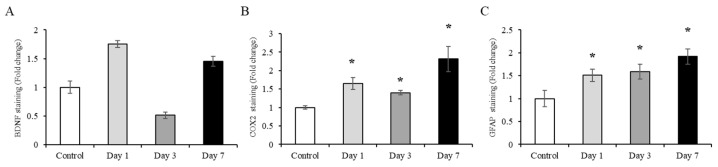
Quantification of acute severe-stress-induced changes in neuroinflammation marker expression in the hippocampus. Acute severe-stress-altered neuroinflammation marker expression in the hippocampus. Changes in the expression of brain-derived neurotrophic factor (BDNF), cyclooxygenase-2 (COX2), and glial fibrillary acidic protein (GFAP) were quantified using ImageJ: (**A**) BDNF; (**B**) COX2; and (**C**) GFAP expression changes in the hippocampus. Data are expressed as means ± SD; * *p* < 0.05, *n* = 6 per group.

**Figure 4 ijms-22-09960-f004:**
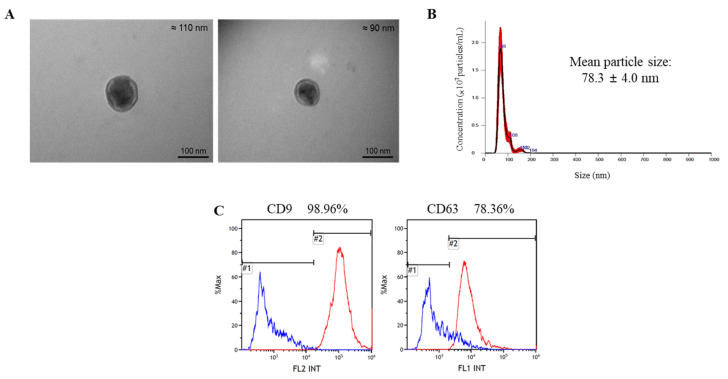
Characterization of total exosomes (tEVs) isolated from serum: (**A**) Transmission electron microscopy (TEM) images showing exosome morphology and size. Scale bar: 100 nm; (**B**) Nanoparticle tracking analysis (NTA) of exosomes to confirm size distribution; (**C**) Flow cytometry (FACS) data confirming the detection of exosome markers (CD9 and CD63).

**Figure 5 ijms-22-09960-f005:**
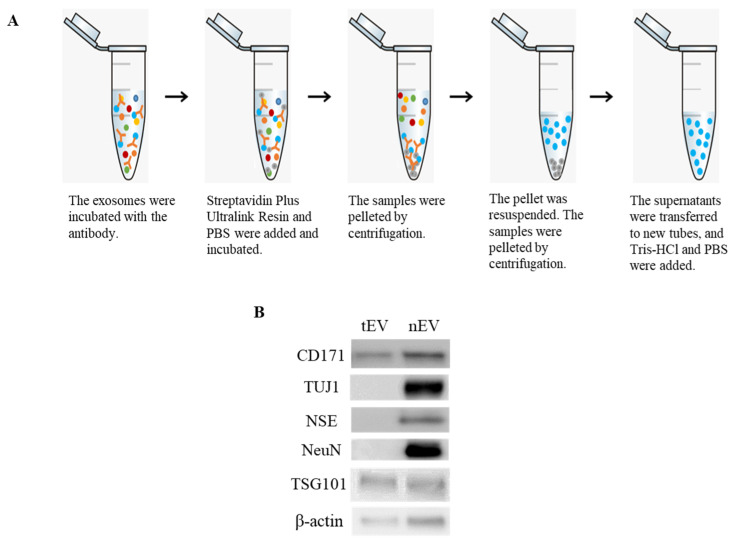
Isolation and characterization of neuronal exosomes (nEVs) in serum: (**A**) Protocol for the isolation of nEVs; (**B**) Western blot images showing the enrichment of nEV-associated markers (nEV marker: CD171; neuronal markers: neuron-specific class III beta-tubulin (TUJ1), neuron-specific enolase (NSE), and neuronal nuclei (NeuN); exosome marker: tumor susceptibility gene 101 (TSG101)).

**Figure 6 ijms-22-09960-f006:**
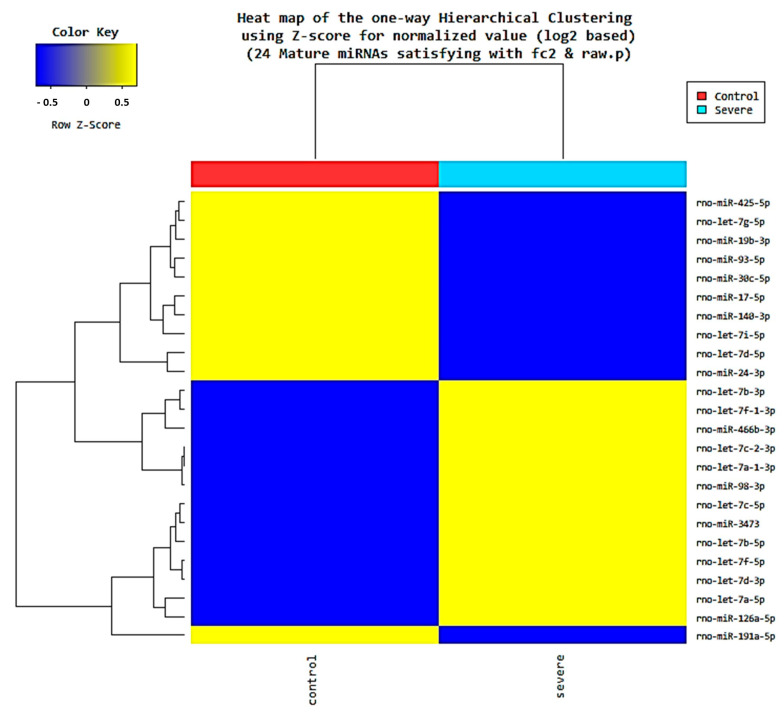
Heatmap of differential miRNA expression in neuronal exosomes (nEVs) induced by acute severe stress. In total, 13 miRNAs were upregulated and 11 miRNAs were downregulated in the Day 7 group compared with the control group.

**Figure 7 ijms-22-09960-f007:**
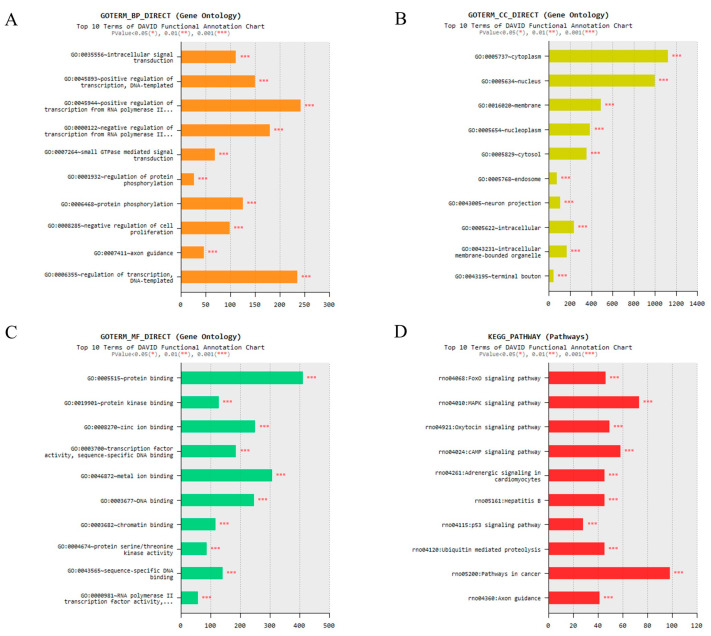
Gene ontology (GO) analysis and Kyoto Encyclopedia of Genes and Genomes (KEGG) pathway analysis indicating that neuronal exosomal miRNAs with different expression patterns under acute severe stress regulate neuron-related genes. (**A**–**C**) Bar graph represents the top 10 GO terms in the three categories: biological process (BP), cell component (CC), and molecular function (MF). (**D**) The most significant 10 KEGG pathways are displayed. * *p* < 0.05, ** *p* < 0.01 and *** *p* < 0.001.

**Figure 8 ijms-22-09960-f008:**
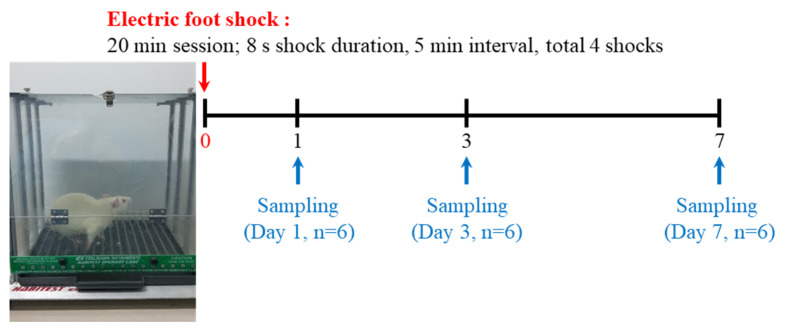
Study design for the induction of acute severe stress. After acute severe stress was induced using electric foot shock, groups were sacrificed at 1 day, 3 days, or 7 days after stress exposure.

**Table 1 ijms-22-09960-t001:** Up/downregulation of miRNAs induced by acute severe stress.

Mature miRNA	Predicted Target Genes	Fold Change (Severe Stress/Control)	miRNA Expression in Acute Severe Stress
rno-let-7a-1-3p	697	48.12	Upregulation
rno-let-7a-5p	379	3.05	Upregulation
rno-let-7b-3p	696	8.42	Upregulation
rno-let-7b-5p	383	4.2	Upregulation
rno-let-7c-2-3p	697	48.12	Upregulation
rno-let-7c-5p	379	3.99	Upregulation
rno-let-7d-3p	15	3.3	Upregulation
rno-let-7f-1-3p	696	10.53	Upregulation
rno-let-7f-5p	380	3.51	Upregulation
rno-miR-126a-5p	389	10.99	Upregulation
rno-miR-3473	114	3.23	Upregulation
rno-miR-466b-3p	487	6.62	Upregulation
rno-miR-98-3p	696	54.13	Upregulation
rno-let-7d-5p	352	0.32	Downregulation
rno-let-7g-5p	382	0.19	Downregulation
rno-let-7i-5p	382	0.11	Downregulation
rno-miR-140-3p	343	0.06	Downregulation
rno-miR-17-5p	525	0.04	Downregulation
rno-miR-191a-5p	53	0.23	Downregulation
rno-miR-19b-3p	485	0.29	Downregulation
rno-miR-24-3p	443	0.18	Downregulation
rno-miR-30c-5p	980	0.45	Downregulation
rno-miR-425-5p	186	0.27	Downregulation
rno-miR-93-5p	520	0.27	Downregulation

## Data Availability

The data that support the findings of this study are available from the corresponding author upon reasonable request.

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
