# Peer review of "Serum-Derived Neuronal Exosomal miRNAs as Biomarkers of Acute Severe Stress"

_ijms, 2021, doi:10.3390/ijms22189960_

Round 1
Reviewer 1 Report
In this interesting study, the authors analyzed miRNAs in serum-derived neuronal exosomes (nEVs) to be used as biomarkers of acute stress. Overall this is valuable study, in which the authors characterize the acute severe stress induced in Sprague–Dawley rats via electric foot shock treatment. In addition, the authors performed NGS to compare neuronal exosomal miRNA at day 7 with the neuronal exosomal miRNA of the control group, and detected 13 upregulated and 11 downregulated miRNAs. The authors suggested that these data provide novel insights into potential stress-related biomarkers. The subject of the article is noteworthy, adding new data in the field of identifying new potential markers to help predicting acute severe stress. The study has an appropriate experimental design, data presentation and discussion, has a good statistical analysis to support the conclusions.
Minor comments:
- I would suggest presenting the expression changes (fold-change or percent-change) of the measured parameters compared to the control group presented at pages 3-4, rather than simply repeating all the data included in the Figures 1 and 3.
- In the Figures’ legends, the authors specify the number of rats per time point and should mention the statistical method used to obtain the p values and the reference group (control group, in this case). Also, the authors must specify the total number of animals used in this study and their number sacrificed at each time point to obtain the brain sections (see Figures 2-3), in the Materials and Methods section (page 10, line 288).
- Given the consecutive data (time) points for the measured parameters (i.e. cortisol and corticosterone), I would suggest minimally performing OneWay Anova test for the parameter’s timing trend, and not only the T-test for each time point vs control.
- Following the NGS analysis, I would suggest performing a brief bioinformatics analysis (using online tools like mirWalk, David or others) to identify the relevant biological processes altered by the differentially expressed miRNAs, as already stated at page 10, line 262-264. Therefore this paragraph from Discussion could be detailed also in the Results section.
- Table 2 could be moved as Supplementary material, being unnecessary for the main manuscript text; accordingly, Table 1 could be supplemented with the corresponding miRBase number for each miRNAs.
- The authors should specify the blood volume collected from each rat and if this procedure was the final one before sacrificing for brain collection or was a separate procedure (page 11, line 300).
- The authors should provide the software used for statistical analysis and graphical design.
- The authors did not comment about the identified study’s limitations.
- Reference #1 is incomplete, please correct it.
Author Response
1. I would suggest presenting the expression changes (fold-change or percent-change) of the measured parameters compared to the control group presented at pages 3-4, rather than simply repeating all the data included in the Figures 1 and 3.
Answer: Thank you for your comments. The manuscript has been revised. The revised sentence is inserted in lines 99, 103-104, 112, 113, 131-133, 135-136, and 138-140.
2. In the Figures’ legends, the authors specify the number of rats per time point and should mention the statistical method used to obtain the p values and the reference group (control group, in this case). Also, the authors must specify the total number of animals used in this study and their number sacrificed at each time point to obtain the brain sections (see Figures 2-3), in the Materials and Methods section (page 10, line 288).
Answer: Thank you for your comments. The manuscript has been revised. The revised sentence is inserted in lines 109, 155 and 323-324.
3. Given the consecutive data (time) points for the measured parameters (i.e. cortisol and corticosterone), I would suggest minimally performing OneWay Anova test for the parameter’s timing trend, and not only the T-test for each time point vs control.
Answer: Thank you for your comments. We compared the differences in glucocorticoid hormones such as cortisol and corticosterone between the control and the severe stress group. Therefore, analysis was performed between the two groups. In addition, the hormones cortisol and corticosterone are released by the circadian rhythm. In rodents, corticosterone peaked in the evening hours, whereas in humans, cortisol levels peaked in daytime. Therefore, we did not perform ANOVA analysis for all times because we considered the amount of hormone that changes with the circadian rhythm. *(Compr Physiol 2014, 4:1273-1298)
4. Following the NGS analysis, I would suggest performing a brief bioinformatics analysis (using online tools like mirWalk, David or others) to identify the relevant biological processes altered by the differentially expressed miRNAs, as already stated at page 10, line 262-264. Therefore this paragraph from Discussion could be detailed also in the Results section.
Answer: Thanks for your kind comments. I performed bioinformatics analysis according to the recommended online tools mirWalk and DAVID. The manuscript has been revised and DAVID data excel file was added supplementary materials S2. The revised sentence is inserted in lines 199-211, 218-224, and 411-413.
5. Table 2 could be moved as Supplementary material, being unnecessary for the main manuscript text; accordingly, Table 1 could be supplemented with the corresponding miRBase number for each miRNAs.
Answer: Thank you for your comments. Table 2 has been moved to Supplementary material S1. And the manuscript has been revised. The revised sentence is inserted in lines 197.
6. The authors should specify the blood volume collected from each rat and if this procedure was the final one before sacrificing for brain collection or was a separate procedure (page 11, line 300).
Answer: Thank you for your comments. The manuscript has been revised. The revised sentence is inserted in lines 326.
7. The authors should provide the software used for statistical analysis and graphical design.
Answer: Thank you for your comments. Thank you for your comments. We added information about statistical analysis software on statistical analysis section. The revised sentence is inserted in lines 416.
8. The authors did not comment about the identified study’s limitations.
Answer: Thank you for your comments. This study’s limitation was mentioned in lines 306-309. And other limitation was added to the manuscript. The revised sentence is inserted in lines 309-311.
9. Reference #1 is incomplete, please correct it.
Answer: Thank you for your comments. The manuscript has been revised. The manuscript has been revised. The revised sentence is inserted in lines 435-436.
Reviewer 2 Report
In this article, entitled “Serum-derived neuronal exosomal miRNAs as biomarkers of acute severe stress”, the authors tested whether the exosomal miRNAs can be stress-specific biomarkers. The authors characterized neuronal exosomes by using a well-established acute severe stress model (foot shock test). The authors clearly identified miRNA candidates for potential biomarkers of acute severe stress. Therefore this work provides therapeutic possibilities. It would be great if the authors can reveal whether the exosomal markers are conserved in humans in the future.
Author Response
In this article, entitled “Serum-derived neuronal exosomal miRNAs as biomarkers of acute severe stress”, the authors tested whether the exosomal miRNAs can be stress-specific biomarkers. The authors characterized neuronal exosomes by using a well-established acute severe stress model (foot shock test). The authors clearly identified miRNA candidates for potential biomarkers of acute severe stress. Therefore this work provides therapeutic possibilities. It would be great if the authors can reveal whether the exosomal markers are conserved in humans in the future.
Answer: Thank you for your comments. Further studies are currently being planned to confirm the neuronal exosomal markers and their applicability to human stress biomarkers. We hope that the results of this study will help find stress biomarkers in humans.